# Non-Destructive Removal of Dental Implant by Using the Cryogenic Method

**DOI:** 10.3390/medicina58070849

**Published:** 2022-06-25

**Authors:** Burak AK, Emre Gürkan Eroğlu, Abdullah Seckin Ertugrul, Ayla Batu Öztürk, Şakir Necat Yılmaz

**Affiliations:** 1Periodontology Department, Faculty of Dentistry, Mersin University, 33343 Mersin, Turkey; 2Periodontology Department, Faculty of Dentistry, Izmir Katip Çelebi University, 35620 Izmir, Turkey; emregurkan.eroglu@ikc.edu.tr (E.G.E.); abdullahseckin.ertugrul@ikcu.edu.tr (A.S.E.); 3Department of Histology and Embryology, School of Medicine, Mersin University, 33343 Mersin, Turkey; abatuozturk@mersin.edu.tr (A.B.Ö.); nyilmaz@mersin.edu.tr (Ş.N.Y.)

**Keywords:** implant removal, carbon dioxide, cryotherapy, rabbit, tibia, reverse torque

## Abstract

*Background and Objectives*: The gold standard for a successful prosthetic approach is the osseointegration of an implant. However, this integration can be a problem in cases where the implant needs to be removed. Removing the implant with minimal damage to the surrounding tissues is important. Osteocytes cannot survive below −2 °C, but epithelial cells, fibroblasts, and other surrounding tissue cells can. Remodeling can be triggered by cryotherapy at temperatures that specifically affect osteocyte necrosis. In this study, we aimed to develop a method for reversing the osseointegration mechanism and for protecting the surrounding tissues by bone remodeling induced by CO_2_ cryotherapy. *Materials and Methods*: In this study, eight 2.8 mm diameter, one-piece mini implants were used in New Zealand rabbit tibias. Two control and six implants were tested in this study. After 2 months of osseointegration, a reverse torque force method was used to remove all osseointegrated implants at 5, 10, 20, and 30 Ncm. The osseointegration of the implants was proven by periotest measurements. Changes in bone tissue were examined in histological sections stained with toluidine blue after rabbit sacrifice. The number of lacunae with osteocyte, empty lacunae, and lacunae greater than 5 µm and the osteon number in a 10,000 µm^2^ area were calculated. Cryotherapy was applied to the test implants for 1 min, 2 min, and 5 min. Three implants were subjected to cryotherapy at −40 °C, and the other implants were subjected to cryotherapy at −80 °C. *Results*: Empty lacunae, filled osteocytes, lacunae >5 µm, and the osteon count around the implant applied at −40 °C were not significantly different from the control implants. The application of −40 °C for 1 min was found to cause minimal damage to the bone cells. The implants, which were applied for 1 min and 2 min, were successfully explanted on the 2nd day with the 5 Ncm reverse torque method. Test implants, which were applied cold for 5 min, were explanted on day 1. Tissue damage was detected in all test groups at −80 °C. *Conclusions*: The method of removing implants with cryotherapy was found to be successful in −40 °C freeze–thaw cycles applied three times for 1 min. To prove implant removal with cryotherapy, more implant trials should be conducted.

## 1. Introduction

Edentulous regions are often treated with dental implant applications. Implants may need to be removed from the bone due to non-osseointegration, improper positioning, or various mechanical problems [1]. In the removal of the implant, the reverse torque method is applied first [2]. While this process is extremely easy for partially osseointegrated implants, it may become impossible if the implant is highly osseointegrated into the bone. In this case, the implants can be explanted by removing the bone around the implant [3]. In patients with chronic diseases that adversely affect bone metabolism but require removal of the implant, removal should be performed with minimal damage to the surrounding tissues [4].

Implant stability can be defined as being primary or secondary. The physical shape of the implant, the structure of the bone, and the surgical technique used are effective in primary stability [2,5,6]. In secondary stability, the biological interaction of the bone with the implant surface is a determining factor [7]. During the transition from primary to secondary stability, the bone undergoes remodeling. Due to remodeling of the bone tissue, implant stability is reduced during a particular period. Fibrotic healing occurs if the implant is put into function in this period, and osseointegration may fail [8,9]. Various methods have been studied for implant removal using bone metabolism. These studies used the decrease in stability during the remodeling of necrotic tissue caused by lasers, electrocautery devices, or direct heating devices within the osseointegrated area [10,11,12]. However, these methods damage the surrounding tissue and intercellular substances along with the bone [13].

Heat-induced thermal necrosis may also be accomplished at cold temperatures. When −20 °C is reached at the bone, crystallization and bone necrosis occurs. The lesion can then heal completely by activating bone formation and by removing necrotic tissue [14].

The highest temperature value at which the osteocyte is affected by cryotherapy is −2 °C. At temperatures below −2 °C, the osteocyte cannot survive. However, keratinocytes, endothelial cells, and fibroblasts continue to live [15]. Cooling the bone below −2 °C selectively affects osteocytes. In this way, the viability of other cells is preserved and a period of reduced secondary stability can be established. By cooling the implant to cryogenic temperatures, heat can be conducted to the bone and necrosis may occur in the regional bone tissue in contact with the implant surface [16].

Thus, a temporary stability reduction occurs during the induced remodeling. In this case, the implant can be removed with a low reverse torque force.

Implant removal methods are generally classified into reverse torque methods and bone removal methods [17]. The reverse torque force method has a high success rate and is minimally invasive, compared with bone removal methods [4,18]. If the reverse torque force is greater than 200 Ncm, or there is more than 4 mm of osseointegrated bone, bone removal methods around the implant are recommended to prevent implant fracture [4]. It has also been reported that implants can fracture at values below 35 Ncm in the reverse torque method [17,19]. Implant removal torque is defined as the torque required to remove the implant [20]. To avoid the complications related to implant removal, it is important to reduce the implant removal torque.

The hypothesis of this study is that, by cooling the implant below −2 °C, local bone necrosis can occur and the surrounding tissues can be protected. By triggering remodeling with cryotherapy, the implant is able to be removed with a low implant removal torque during the period of reduced secondary stability.

This study aims to determine a minimally invasive implant removal protocol at cryogenic temperatures.

## 2. Materials and Methods

### 2.1. Surgical Procedures

Ethical approval for this study was obtained from the Mersin University Experimental Animals Ethics Committee (approval number: 52602694-050.01.04-E899168). One New Zealand white rabbit was included in the study. The rabbit was injected intramuscularly with ketamine hydrochloride (Ketamidor-Richet Pharma AG, Wels, Austria) (35 mg/kg) and xylazine hydrochloride (Rompun 2%, Bayer, Istanbul, Turkey) (5 mg/kg) as anesthesia, following the rules of asepsis and antisepsis. The operation area was exposed and covered with sterile drapes. For local hemostasis, infiltrative local anesthesia was applied to the operation area with 4% articaine (Ultracain D-S Forte-Aventis, Istanbul, Turkey) containing 0.5 cc, 0.006 mg/mL epinephrine. The rabbit tibia was disinfected with an iodine solution. Then, an anterior–posterior incision was made with No. 15 scalpel and the full-thickness flap was reflected. The implant bed site along the midline of the tibia was exposed. Using retractors, a sufficient field of vision was created in the surgical site, and a one-piece implant with an outer diameter of 2.8 mm (Mini Sky Implant System, Bredent Medical GmbH & Co., Senden, Germany) was placed with placement drills, as recommended by the manufacturer, via sterile saline irrigation. It was placed at a distance of 3 mm between the two implants. The operation area was closed with 3.0 vicryl sutures (Ethicon, Somerville, NJ, USA). After the surgical procedures were completed, the rabbit was followed in a Eurostandard Type IV cage under normal care conditions.

### 2.2. Cryotherapy Procedures

A custom female socket compatible with the implant abutment was made on the cryoprobe with the CNC manufacturing method (Figure 1). In this way, standardization was achieved in cold application. The cold application was made using a Cryoprobe (Universal Cryo-Unit Inc., Ankara, Turkey). A −40 °C cryoprobe was used for right tibia implants, and a −80 °C cryoprobe was used for left tibia implants.

Three freeze cycles were applied to four implants in each tibia (Mini Sky implant System, Bredent Medical GmbH & Co., Senden, Germany): control, 1 min, 2 min, and 5 min, respectively. After each cryotherapy application, we waited 5 min, defined as the thaw cycle.

### 2.3. Implant Stability Measurements

After two months of osseointegration, periotest (Periotest, Medizintechnik Gulden, Modautal, Germany) measurements were performed before cryotherapy application. The periotest measurements were obtained by positioning the tip of the instrument perpendicular to the axis of the implant, as described by the manufacturer. Three measurements per implant were obtained, and the average of these recordings was calculated as (Periotest values) PTV.

### 2.4. Implant Removal Procedures

Removal of the implant with the reverse torque force method was attempted with a physiodispenser (Saeshin Traus, Daegu, Korea) at torque settings of 5, 10, 20, and 30 Ncm. Implant removal trials were performed at the same time each day. Each implant was tested with the reverse torque method for proof of osseointegration prior to cryotherapy. Reverse torque was performed immediately after cryotherapy. The reverse torque method was repeated on the 1st and 2nd days. Attempts were made to determine the day on which osseodisintegration started.

### 2.5. Histological Examination

After successful explantation of the implants, the rabbit was sacrificed. The samples were fixed in formalin and decalcified with a 10% Ethylenediaminetetraacetic acid (EDTA) solution. Then, the samples were embedded in paraffin. Cross sections (5 µm thick) were cut using a microtome (Leica 2125RT) and mounted on glass slides. Following deparaffinization in xylene and rehydration in graded ethanol series, the sections were stained with 4% toluidine blue. Photomicrographs were taken using an Olympus microscope equipped with an Olympus LC30 digital camera (Olympus Optical Company Corp., Tokyo, Japan).

By dividing each implant area into three areas—coronal, middle, and apical—six sections from each area at different levels were examined. For each section, an area of 10,000 µm^2^ was evaluated in four different (north, south, east, and west) regions of the implant socket [7] (Figure 2). The number of lacunae filled with osteocytes, empty lacunae, and lacunae greater than 5 µm (enlarged lacunae) and the number of osteons were calculated (Figure 3).

### 2.6. Statistical Analyses

The effect was evaluated by comparing the number of lacunae with osteocytes, empty lacunae, and lacunae greater than 5 μm and the number of osteons in a 10,000 µm^2^ area. One-way analysis of variance (ANOVA) followed by Tukey’s multiple comparison test was performed using GraphPad Prism version 9.3.1 for Windows, GraphPad Software, San Diego, CA, USA www.graphpad.com accessed on 9 September 2021). A *p*-value < 0.05 was considered significant.

## 3. Results

### 3.1. Histological Examination

A histopathological examination of the effect of the cold application on the implants according to the application time was performed, and the results are presented in Table 1.

For the amount of right empty lacuna, the number of empty lacunae increased as the application time increased. For the number of right filled osteocytes, as the application time increased, the number of filled osteocytes decreased. The number of lacunae larger than 5 µm detected in the right tibia was at the lowest at 1 min and the highest at 5 min. The number of osteons in the right tibia decreased as the application time increased. The number of empty lacunae on the left was always lower than on the right, and this number increased as the application time increased. For the left filled osteocyte counts, unlike on the right side, the number of lacunae of filled osteocytes increased at 5 min. Except for the 5 min application, the number of lacunae decreased as the application time increased. The number of lacunae larger than 5 µm tended to increase as the application time increased. The left osteon count tended to decrease with increasing application time.

In the comparison within the group, no significant differences were found in the number of empty lacunae in the right tibia between the control and 1 min, and between 1 min and 2 min. A significant difference was found between the control, and 2 min and 5 min. A significant difference was found between 1 min and 5 min, and between 2 and 5 min. In the comparison within the group, a significant difference was found in the number of empty lacunae in the left tibia between the control, and 1 min, 2 min, and 5 min. No significant differences were found among 1 min, 2 min, and 5 min groups. (Figure 4 and Table 2).

No significant differences were found in the number of filled osteocyte lacunae in the right tibia between 1 min and the control, between 2 min and the control, and between 1 min and 2 min. Significant differences were found between the control and 5 min, between 1 min and 5 min, and between 2 min and 5 min. No significant differences were found in the number of lacunae filled with osteocytes in the left tibia between the control, and 1 min, 2 min, and 5 min. No significant differences were found among 1 min, 2 min, and 5 min groups (Figure 5 and Table 3).

No significant differences were found between the control and 1 min in the number of lacunae larger than 5 µm in the right tibia. A significant difference was found between 2 min and the control, between 5 min and the control, between 1 min and 2 min, between 1 min and 5 min, and between 2 min and 5 min. The number of lacunae greater than 5 µm in the left tibia was not significantly different between the control, and 1 min and 2 min. A significant difference was found between the control and 5 min. No significant difference was found between 1 min and 2 min. A significant difference was found between 1 min and 5 min, and between 2 min and 5 min (Figure 6 and Table 4).

No significant differences were found in the number of right osteons between the control and 1 min, and between 1 min and 2 min. A significant difference was found between the control and 2 min, and between the control and 5 min. No significant differences were found between the number of osteons in the left tibia at 1 min and between control and 5 min. A significant difference was found between the control and 2 min. No significant difference was found between 1 min and 5 min. A significant difference was found between 1 min and 2 min, and between 2 and 5 min (Figure 7 and Table 5).

In the comparison between the control implants, no significant differences were found between the number of empty lacunae, the number of lacunae filled with osteocytes, the number of lacunae greater than 5 µm, and the number of osteons (Figure 8 and Table 6).

The number of empty lacunae and osteocyte-filled lacunae numbers in the right and left tibia was not significantly different in the implants that were applied for 1 min. The number of lacunae greater than 5 µm was significantly different between the right and left tibia. The number of osteons was significantly different between the right and left tibia (Figure 9 and Table 7).

The number of empty lacunae in the right and left tibia was significantly different in implants that were applied for 2 min. The number of osteocyte-filled lacunae and the number of lacunae larger than 5 µm were not significantly different between the right and left tibia. The number of osteons was significantly different between the right and left tibia (Figure 10 and Table 8).

The number of empty lacunae in the right and left tibia was significantly different in the implants applied for 5 min. The number of osteocyte-filled lacunae was significantly different between the right and left tibia. The number of lacunae greater than 5 µm in the right and left tibia was not significantly different. The number of osteons was significantly different between the right and left tibia (Figure 11 and Table 9).

### 3.2. Periotest Measurement

Implant osseointegration was evaluated 2 months after implantation. Osseointegration of the implants tested with periotest measurements is presented in Table 10**.**

Osseointegration was assessed on a scale ranging from −8 (very stable) to +50 (extremely unstable) in the periotest measurements [21]. In this study, implant stability was determined to be always high in the periotest measurements.

## 4. Discussion

In this study, we developed a method for non-destructive implant explantation on the osseointegrated implants by inducing selective tissue destruction at cryogenic temperatures.

In dental implant applications, implant explantation may be required due to mechanical or biological complications [1]. In these cases, most implants are well osseointegrated and the explantation of the implants can be both invasive and damaging to the surrounding tissues [4,22]. Considering criteria such as adjacent anatomical structures, existing bone, bone quality, and mobility when providing clinical advice for implant explantation, the reverse torque force method is recommended first and then osteotomy with trephine burs or piezoelectric surgery is recommended [4]. The most commonly used implant removal method is the reverse torque force method. However, the success rate of this method is 87.7% [18]. The second method is explantation with trephine burs [18]. For the success of the trephine bur method, at least 1.5 mm of the cortical bone should be around the implant. A larger implant also needs to be placed in the removed area [1]. In cases where the implant cannot be removed, the recommended methods are invasive procedures [10]. The hypothesis of reversing the biological processes of osseointegration using heat for implant removal is an innovative approach.

Heat-induced necrosis of bone is avoided due to surgical principles [23]. The threshold for bone viability has been reported to be 47 °C [24,25]. However, in cases where a temperature increase is not considered in implantation processes, osseointegration cannot be completed and implant loss occurs [9,13]. It has been found that 60 °C for a 1 min application reduces the bone implant contact (BIC) rate and causes crestal bone loss around the implant [24,26]. Implant removal trials have been carried out with lasers and electrocautery to generate heat above 47 °C around the implant [10,12]. The heat applied in these studies causes histological damage to the bone’s cells and inorganic matrix [27].

Implant osseodisintegration devices using hot and cold applications have been patented. However, no articles other than case reports about the clinical use of these devices have been published in PubMed and MEDLINE [28]. Thus, this study, in which in vivo osseodisintegration was applied within cryogenic temperatures, is unique in the literature.

Cryotherapy application preserves the inorganic matrix while performing cell death in the tissue [29]. Since no change is observed in the inorganic matrix, changes in the BIC values of the implants may not be expected in the short term. After cryotherapy application, cell death occurs, but minimal tissue loss occurs as the tissue matrix is not damaged. This matrix should act as a scaffold for remodeling [27].

Periotest values (PTV) provide measurable data about the osseointegration between bone and implant. It can be loaded immediately if the PTV value is negative, and positive values are a contraindication for loading [21]. In this study, periotest measurements were used to monitor whether the implants were osseointegrated.

No significant change in PTV values after cryotherapy was observed. All of the implants were osseointegrated in this study. This may be explained by the fact that cryotherapy does not alter the inorganic matrix [27]. All tested implants were removed with a force of 5 Ncm at the end of the experiment. Remodeling of the bone tissue adjacent to the implant may have reduced the resistance to the reverse torque force [8,30].

Implants are most often removed with the reverse torque force method [4]. This method can be successful if the fracture toughness of the implant material is high [4,28]. Zirconium implants have a fracture toughness of 4-18 MPa/m, and titanium implants have a fracture toughness of 77 MPa/m [4,31,32]. Since zirconia implants have lower fracture toughness values than titanium, they cannot be explanted by the reverse torque force method [4].

The thermal conductivity of bone is lower than titanium. The thermal conductivity of bone is 0.54 ± 0.020 W/mK [33]. The thermal conductivity of ceramics is similar to titanium. According to the American Society for Testing and Materials (ASTM International), the thermal conductivity of titanium alloys is between 20.1 and 22.6 W/mK [34]. Zirconium is 20.5 W/mK [35]. In hot or cold applications, the diffusion of energy to the entire implant surface can be expected to be faster than its transmission to the bone. In this study, the duration of cryotherapy for implant explantation was determined.

Several problems in the explantation of ceramic implants exist. Heat generation around titanium dental implants has been studied to remove the implant by thermal necrosis [11,25,36]. Local temperature increase around the implant was investigated using electrocautery [12]. Due to the electrical conductivity of titanium in implants, working with electrocauteries is possible. The electrical conductivity of zirconium is lower than titanium [37]. Therefore, zirconium implants cannot be removed by reverse torque or electrocautery. Only osteotomy can be performed in implant removal [4]. Therefore, we expected that the biological process occurring in titanium implants after cryotherapy will also occur in ceramic implants. Cryotherapy-assisted removal tests are also required for ceramic implants.

Implant removal with heat generation around the implant using lasers has also been studied. In titanium implants, a temperature increase of 10 °C was noticed at 4 W in 17 s for the diode laser and 15 s for the Er:YAG (erbium-doped yttrium aluminium garnet) laser [34]. Based on this calculation, the time required for the osseointegrated implant to reach the thermal threshold of 47 °C from 36 °C in bone was less than 1 min.

A human dental implant was removed by inducing thermal necrosis with a CO_2_ laser. The researchers applied the laser by waiting for 40 s after 40 s of application and increasing it from 4 Watts to 6 Watts at four intervals. Heat at 70 °C was generated on the implant surface. One week later, the implant was removed with a 37 Ncm reverse torque [10].

Implants have also been reported to be able to be removed via thermal necrosis based on finite element analysis modeling. Controlled necrosis has been reported to be able to be performed at low electrocautery values, and the size of the implant and the diameter of the electrocautery tip are important parameters. Some researchers have recommended using wide tips due to the slow heat increase and have recommended using short-term contacts in small implants [11].

In vivo, Wilcox et al. applied 5-Watt unipolar electrocautery to the implants for 1 s and reported a temperature increase of 8.87 °C [12]. In another in vivo study, mono-polar electrocautery was used, and the most effective method was reported to be 40 Watt for 40 s. A reverse torque force of 27.9 ± 12.1 Ncm was measured in the control group, and a reverse torque force of 18.4 ± 6.59 Ncm was measured in the test group. Histological examination or clinical use of the implant area removed via electrocautery has not yet been reported in these studies. However, 60 °C for 1 min application during an implantation process has been determined to reduce the BIC rate and to cause crestal bone loss around the implant bed site [26]. Crestal bone loss may occur in humans after cryotherapy.

Thermal necrosis can also occur at negative temperatures [38]. Bone cells cannot survive at temperatures below −2 °C. However, endothelial cells can survive from −15 °C to −40 °C, and fibroblasts can survive from −30 °C to −35 °C [15]. While the bone cells are selectively affected in the cryogenic temperature range, damaging the fibroblast and epithelial cells is not possible. Additionally, unlike at high temperatures, the inorganic matrix is not thermally damaged [27]. When the implant’s heat conduction and the bone’s heat conduction are calculated, the short-term low temperature only damages the implant–bone contact area, and the surrounding tissues are protected. With the initiation of the remodeling mechanism, osteoid tissue will form around the implant [8,30]. A temporary decrease in secondary stability can be expected [9]. In this way, removing the implant from the bone using the biological process is possible.

In vitro, when a 2 mm cryoprobe was applied to the rabbit tibia, at an application temperature of −150 °C, −30 °C was reached in 1 min at a distance of 4 mm [16]. We speculate that the application of −80 °C affects less bone tissue in 1 min. Additionally, fewer pathologic fractures and fewer infections were reported at −50 °C than at −196 °C [29]. The safe temperature in bone tumors has been reported as −30 to −40 °C and approximately 1 min of application [39]. For this reason, −80 °C and −40 °C CO_2_ cryotherapy were preferred in this study.

Multiple repetitions of rapid cooling and slow heating are recommended to achieve maximum cell death in the target organ tissue with cryotherapy [40]. For the application of cryotherapy in the maxillofacial region, the recommendation is to apply a 1 min application and a 5 min resolution cycle for lesions up to 2 cm three times. For lesions larger than 2 cm, it is recommended to apply cryospray directly to the lesion for 1 min after resection and perform a 5 min thaw cycle twice [41]. Since a 1 min application time is frequently recommended in studies, 1 and 2 min application times were planned for the test. The 5 min application was planned as a positive control in the test.

Based on the above information, this study investigated the effects of cryogenic temperatures on bone histology, osseointegration, and reverse torque strength for implant removal. Eight implants were placed in the rabbit’s tibia. The 60-day osseointegration period was followed [42]. After checking the osseointegration with seven group combinations, a temperature of −80 °C was applied to the right tibia and, −40 °C was applied to the left tibia implants. Cryotherapy was applied to the implants three times as 0 min (control), 1 min, 2 min, and 5 min freezing, as well as 5 min thawing cycles. Implant removal was attempted with a reduced reverse torque force.

According to the finite element analysis, the implant with a relative osseointegrated area (ROA) of 2% can withstand a reverse torque of 30 Ncm, while the implant with a 100% ROA can withstand a reverse torque of around 45 Ncm [43]. Since the prosthetic parts are screwed at 35 Ncm [44], an implant planned for clinical use should withstand a minimum torque of 35 Ncm. For this reason, before the cryotherapy application, a reverse torque of 35 Ncm was applied. No movement was observed in any of the implants.

Many implant studies are performed on the tibia of rabbits [45,46]. Therefore, a 2.8 mm implant placement in a rabbit tibia was preferred in the cryogenic application experiment. The shrinkage of the implant at cryogenic temperatures affects the deterioration of the integrity between the implant and the bone [28]. However, the failure to apply the reverse torque method immediately after applying cryotherapy contradicts this notion. No implants up to 35 Ncm could be removed immediately after the cryotherapy application. This suggests that the biological processes are effective rather than the dimensional changes in the implants.

After the cryotherapy application, the implants that were applied for 5 min on the 1st day could be removed with a torque of 5 Ncm at both −40 °C and −80 °C. All other implants tested were able to be removed on day 2 with a reverse torque of 5 Ncm. A control implant could not be removed with the reverse torque force method. The other control implant could be removed at 40 Ncm.

In the literature, cryotherapy has been used in dentistry to treat locally aggressive tumors [29]. Limited information could be obtained due to the lack of similar in vivo studies using cryotherapy for implant removal. Our study results determined a significant relationship between the application time and bone viability at cryogenic temperatures. When the empty lacuna data between the control and test groups were examined, a significant relationship was found between the control, and 2 min and 5 min for the −40 °C application. However, no significant relationship was found between the control and the 1 min treatment. The fact that 40 °C for 1 min did not increase the number of empty lacunae suggests that it does not impair bone viability. No significant differences were found in the number of lacunae filled with osteocytes between the control, and the 1 min and 2 min treatments. A significant difference was found between the control and 5 min. However, the 1 min and 2 min durations did not impair bone viability. The number of lacunae greater than 5 µm was not significant between the control and 1 min. However, it differed significantly between 2 and 5 min, and the control. This difference may suggest a lack of effect on bone viability in cases where 1 min of application is made. No significant relationship was found between the control and 1 min in the number of osteons in the tibia with −40 °C application. However, a significant relationship between the control, and 2 min and 5 min was found. In this case, 1 min of application did not impair the viability of the bone.

In the application of −80 °C, the bone cells and surrounding tissues are expected to be damaged. However, the core temperature of the bone was predicted to not decrease to −80 °C due to the blood flow in the bone, the low heat conduction of the bone, and the limited time of application [47]. For this reason, the effect of a −80 °C application was also evaluated. At these temperatures, bone viability was more negatively affected. The number of empty lacunae was significantly different between the control and 1 min and significantly different between 2 min and 5 min. However, no significant relationship was found between the control, and 1 min, 2 min, and 5 min in the number of lacunae filled with osteocytes. In the number of lacunae larger than 5 µm, no significant relationship was found between the control, and 1 min and 2 min applications. However, a significant relationship was found between the control and 5 min. The number of osteons did not differ significantly between the control, and 1 min and 5 min at −80 °C. However, a significant difference was found between the control and 2 min. All of the control and test groups differed in the number of empty lacunae. No significant difference was found in the number of osteocytes. A difference was found between 5 min and the control in the number of lacunae larger than 5 µm, and a significant difference was found in the osteon count for 2 min of application. A study in which periods of less than 1 min are evaluated in terms of freezing time for the −80 °C application would be more significant.

The most important finding of this study is that a 1 min application at −40 °C may be safe, and less than 1 min should be evaluated at −80 °C. The implants that were applied for 5 min could be removed with a reverse torque of 5 Ncm on the 1st day. However, implants that were applied for 1 min and 2 min on the 2nd day could be removed at 5 Ncm after cryotherapy. According to the parameters evaluated in this study, a 1 min application at −40 °C for non-destructive removal of the implant in the in vivo environment has no effect on the viability of the bone.

A rabbit’s metabolism is faster than human metabolism [27]. In this experiment, the implants were removed with a force of 5 Ncm on the 1st and 2nd days. A human case report revealed that an implant was removed at 37 Ncm after one week with thermal necrosis [10]. At least one week of osseodisintegration is needed in humans at negative temperatures.

Implant removal ability with a reverse torque of 5 Ncm after cryotherapy is advantageous, compared with the ability to remove implants that can be removed at 37 Ncm by creating thermal necrosis at positive temperatures using electrocautery or a laser [10,11].

### Study Limitations and Future Directions for Research in This Field

A 2.8 mm single piece implant was used in this study, but two pieces and different sizes of implants also be tested,A rabbit’s metabolism is faster than that of humans. Further studies should be performed to determine the day of osseodisintegration in humans,A radiofrequency analysis (RFA) could not be performed using the one-piece implant in this study. Studies in which an RFA is conducted on its connection with the bone should be carried out,For ceramic and titanium implants, the cryotherapy explantation test should be repeated in larger trials.

## 5. Conclusions

This study developed a protocol for non-destructive implant removal at cryogenic temperatures,Osseointegrated implants could be removed with a reduced reverse torque force without osteotomy,Osseointegrated implants that could withstand a 35 Ncm reverse torque force could be removed with a 5 Ncm reverse torque force after −40 °C cryotherapy,Three 1 min freeze and 5 min thaw cycles at −40 °C had a minimal impact on bone tissue viability,More studies are needed to fully understand the osseodisintegration of dental implants, and studying the effects of cryogenic temperatures on dental implants and bone metabolism in humans is necessary,Studies examining bone and soft tissue healing after the removal of an implant with cryotherapy should be conducted.

## Figures and Tables

**Figure 1 medicina-58-00849-f001:**
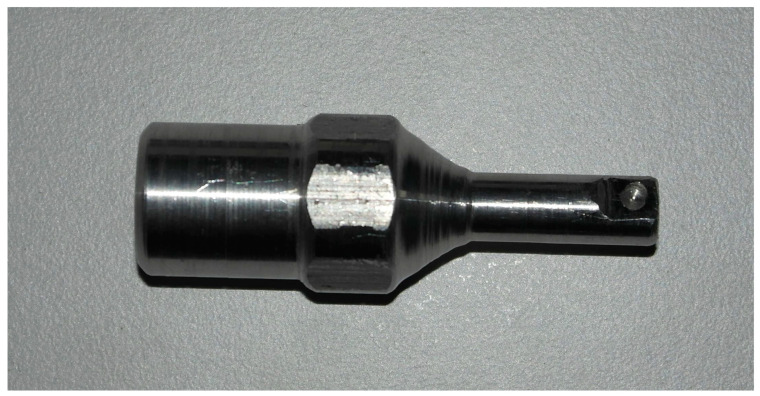
Cryotherapy probe modification.

**Figure 2 medicina-58-00849-f002:**
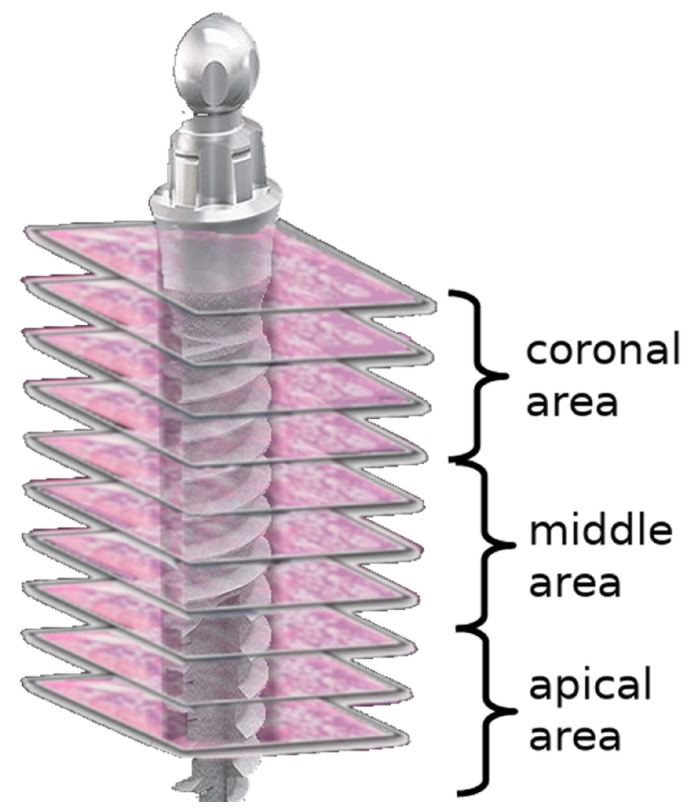
Scheme demonstrating the regions of interest for the evaluation of bone vitality.

**Figure 3 medicina-58-00849-f003:**
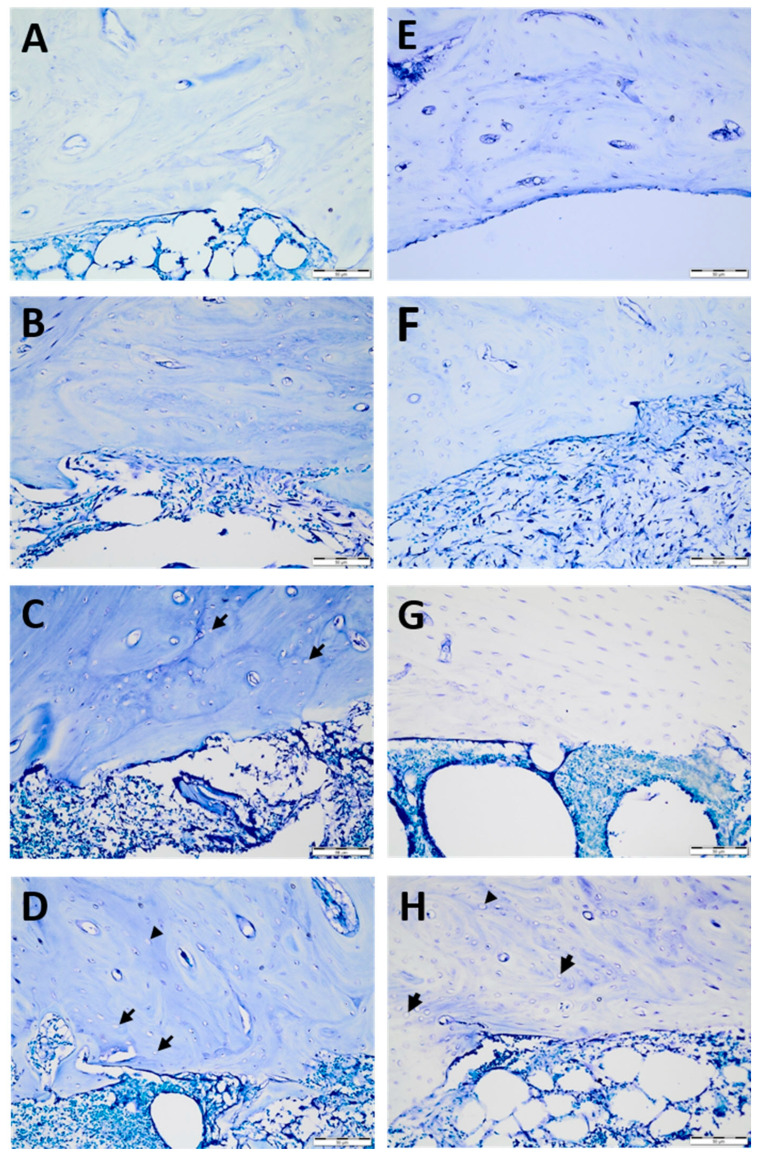
Histological sections of temperature-dependent bone tissue deterioration stained with toluidine blue: −40 °C cryoprobe area (**A**–**D**) for the control (**A**), 1 min (**B**), 2 min (**C**), and 5 min (**D**) and −80 °C cryoprobe area (**E**–**H**) for the control €, 1 min (**F**), 2 min (**G**), and 5 min (**H**). Empty lacunae (arrows) enlarged lacunae (arrowheads). Scale bar: 50 µm. High-resolution tif images of tibias that underwent −40 and −80 degrees cryotherapy are presented as Appendix A, respectively.

**Figure 4 medicina-58-00849-f004:**
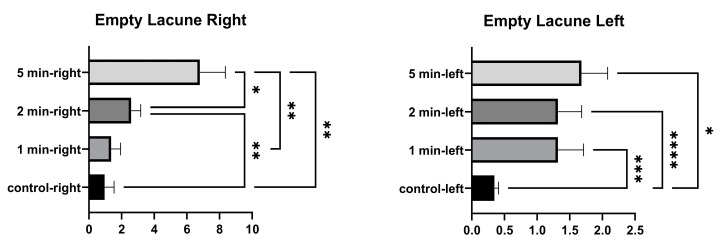
Empty lacunae graphs of the left and right tibia. (*p* ≤ 0.05) as *, (*p* ≤ 0.01) as **, (*p* ≤ 0.001) as *** and (*p* ≤ 0.0001) as ****.

**Figure 5 medicina-58-00849-f005:**
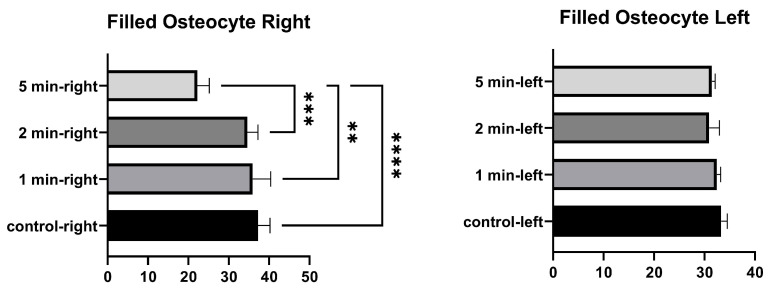
Filled osteocytes graphs of the left and right tibia. (*p* ≤ 0.01) as **, (*p* ≤ 0.001) as *** and (*p* ≤ 0.0001) as ****.

**Figure 6 medicina-58-00849-f006:**
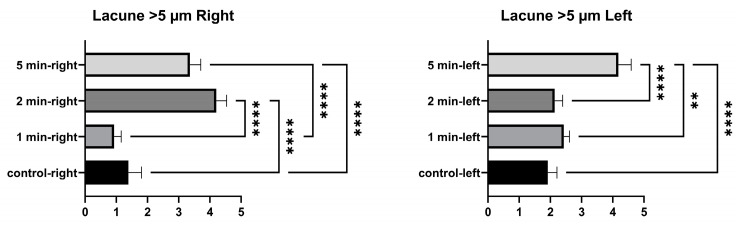
Larger than 5 µm lacunae graphs of the left and right tibia. (*p* ≤ 0.01) as ** and (*p* ≤ 0.0001) as ****.

**Figure 7 medicina-58-00849-f007:**
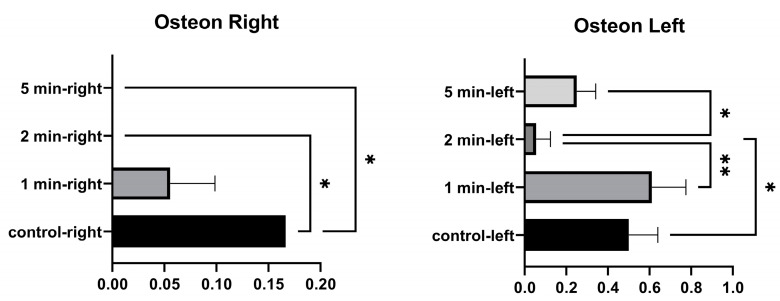
Osteon graphs of the left and right tibia. (*p* ≤ 0.05) as *, (*p* ≤ 0.01) as **.

**Figure 8 medicina-58-00849-f008:**
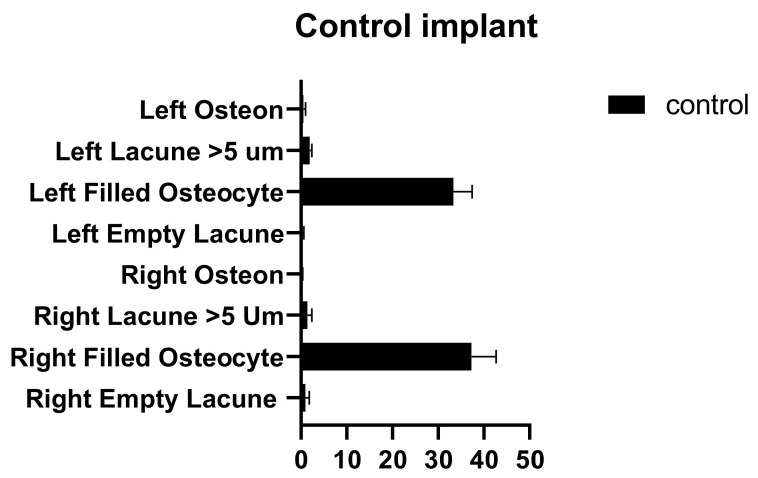
Left and right control implant inter-group comparison graph.

**Figure 9 medicina-58-00849-f009:**
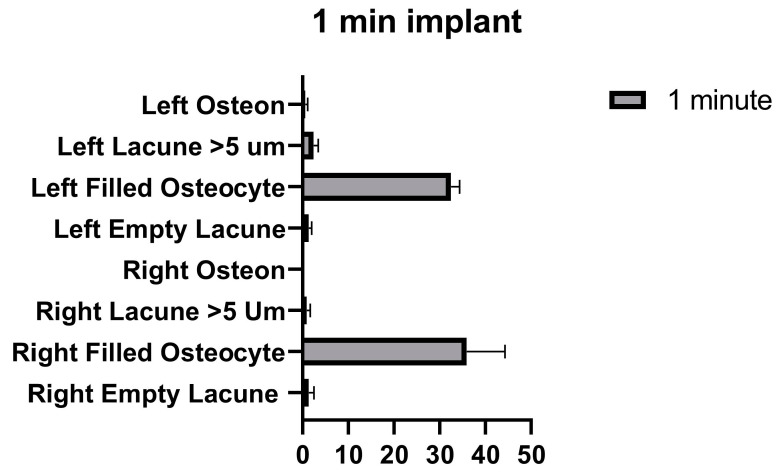
Left and right 1 min implant inter-group comparison graph.

**Figure 10 medicina-58-00849-f010:**
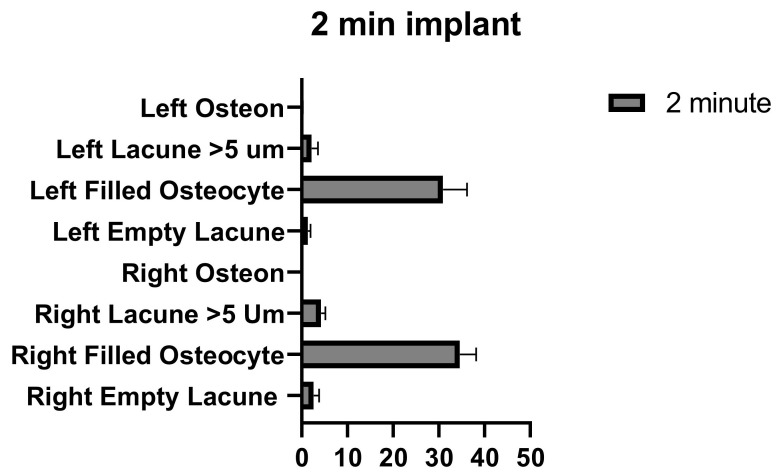
Left and right 2 min implant inter-group comparison graph.

**Figure 11 medicina-58-00849-f011:**
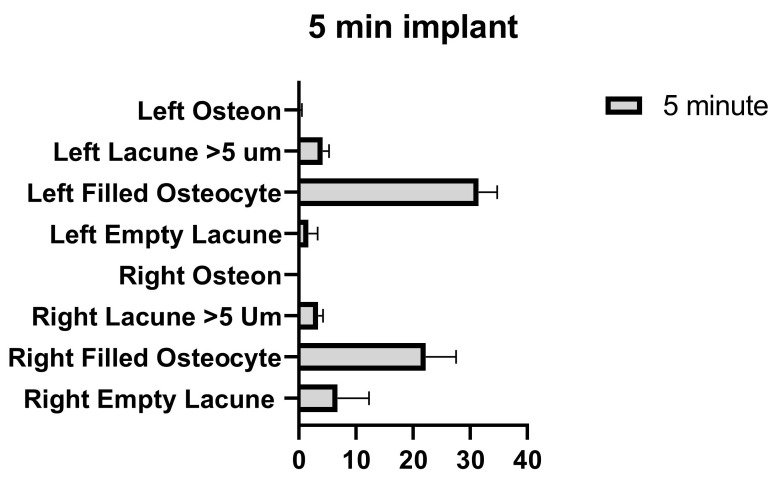
Left and right 5 min implant inter-group comparison graph.

**Table 1 medicina-58-00849-t001:** Empty lacunae, filled osteocytes, lacunae >5 µm, and osteon numbers in a 10,000 µm^2^ area.

Means	Control	1 Min	2 Min	5 Min
Right Empty Lacunae	0.9583	1.361	2.583	6.792
Right Filled Osteocytes	37.28	35.93	34.60	22.24
Right Lacunae > 5 µm	1.389	0.9306	4.208	3.361
Right Osteon	0.1667	0.05556	0.000	0.000
Left Empty Lacunae	0.3472	1.319	1.319	1.681
Left Filled Osteocytes	33.32	32.46	30.90	31.47
Left Lacunae > 5 µm	1.917	2.431	2.139	4.181
Left Osteon	0.5000	0.6111	0.05556	0.2500

**Table 2 medicina-58-00849-t002:** Empty lacunae table of the left and right tibia.

Tukey’s Multiple Comparison Test	Mean Diff.	95.00% CI of Diff.	Summary	Adjusted *p*-Value
Right Empty Lacunae				
Control vs. 1 min	−0.4028	−1.304 to 0.4986	ns	0.5931
Control vs. 2 min	−1.625	−2.575 to −0.6753	***	0.0008
Control vs. 5 min	−5.833	−9.486 to −2,180	**	0.0015
1 min vs. 2 min	−1.222	−2.533 to 0.08900	ns	0.0725
1 min vs. 5 min	−5.431	−9.005 to −1.857	**	0.0024
2 min vs. 5 min	−4.208	−7.578 to −0.8386	*	0.0119
Left Empty Lacunae				
Control vs. 1 min	−0.9722	−1.406 to −0.5389	****	<0.0001
Control vs. 2 min	−0.9722	−1.370 to −0.5746	****	<0.0001
Control vs. 5 min	−1.333	−2.325 to −0.3416	**	0.0068
1 min vs. 2 min	0.000	−0.4413 to 0.4413	ns	>0.9999
1 min vs. 5 min	−0.3611	−1.432 to 0.7094	ns	0.7740
2 min vs. 5 min	−0.3611	−1.319 to 0.5972	ns	0.7111

(*p* > 0.05) summarized as non-significant (ns), (*p* ≤ 0.05) as *, (*p* ≤ 0.01) as **, (*p* ≤ 0.001) as *** and (*p* ≤ 0.0001) as ****.

**Table 3 medicina-58-00849-t003:** Filled osteocytes table of the left and right tibia.

Tukey’s Multiple Comparison Test	Mean Diff.	95.00% CI of Diff.	Summary	Adjusted *p*-Value
Right Filled Osteocytes				
Control vs. 1 min	1.347	−4.199 to 6.894	ns	0.8993
Control vs. 2 min	2.681	−2.074 to 7.435	ns	0.4033
Control vs. 5 min	15.04	9.284 to 20.80	****	<0.0001
1 min vs. 2 min	1.333	−4.959 to 7.625	ns	0.9299
1 min vs. 5 min	13.69	6.037 to 21.35	***	0.0005
2 min vs. 5 min	12.36	6.973 to 17.75	****	<0.0001
Left Filled Osteocytes				
Control vs. 1 min	0.8657	−2.153 to 3.884	ns	0.8465
Control vs. 2 min	2.421	−3.049 to 7.892	ns	0.6003
Control vs. 5 min	1.852	−2.841 to 6.545	ns	0.6816
1 min vs. 2 min	1.556	−1.971 to 5.082	ns	0.6029
1 min vs. 5 min	0.9861	−1.618 to 3.590	ns	0.7080
2 min vs. 5 min	−0.5694	−4.309 to 3.170	ns	0.9720

(*p* > 0.05) summarized as non-significant (ns), (*p* ≤ 0.001) as *** and (*p* ≤ 0.0001) as ****.

**Table 4 medicina-58-00849-t004:** Larger than 5 µm lacunae table of the left and right tibia.

Tukey’s Multiple Comparison Test	Mean Diff.	95.00% CI of Diff.	Summary	Adjusted *p*-Value
Right Lacunae > 5 µm				
Control vs. 1 min	0.4583	−0.3087 to 1.225	ns	0.3547
Control vs. 2 min	−2.819	−3.663 to −1.975	****	<0.0001
Control vs. 5 min	−1.972	−2.627 to −1.317	****	<0.0001
1 min vs. 2 min	−3.278	−4.090 to −2.466	****	<0.0001
1 min vs. 5 min	−2.431	−3.124 to −1.737	****	<0.0001
2 min vs. 5 min	0.8472	0.08690 to 1.608	*	0.0262
Left Lacunae > 5 µm				
Control vs. 1 min	−0.5139	−1.319 to 0.2914	ns	0.3010
Control vs. 2 min	−0.2222	−1.022 to 0.5778	ns	0.8581
Control vs. 5 min	−2.264	−3.044 to −1.484	****	<0.0001
1 min vs. 2 min	0.2917	−1.167 to 1.750	ns	0.9401
1 min vs. 5 min	−1.750	−2.916 to −0.5839	**	0.0027
2 min vs. 5 min	−2.042	−2.919 to −1.164	****	<0.0001

(*p* > 0.05) summarized as non-significant (ns), (*p* ≤ 0.05) as *, (*p* ≤ 0.01) as ** and (*p* ≤ 0.0001) as ****.

**Table 5 medicina-58-00849-t005:** Osteon table of the left and right tibia.

Tukey’s Multiple Comparison Test	Mean Diff.	95.00% CI of Diff.	Summary	Adjusted *p*-Value
Right Osteon				
Control vs. 1 min	0.1111	−0.05363 to 0.2759	ns	0.2579
Control vs. 2 min	0.1667	0.05176 to 0.2816	**	0.0036
Control vs. 5 min	0.1667	0.05176 to 0.2816	**	0.0036
1 min vs. 2 min	0.05556	−0.01610 to 0.1272	ns	0.1620
1 min vs. 5 min	0.05556	−0.01610 to 0.1272	ns	0.1620
2 min vs. 5 min	0.000			
Left Osteon				
Control vs. 1 min	−0.1111	−0.4975 to 0.2752	ns	0.8454
Control vs. 2 min	0.4444	0.1269 to 0.7620	**	0.0049
Control vs. 5 min	0.2500	−0.02553 to 0.5255	ns	0.0828
1 min vs. 2 min	0.5556	0.2540 to 0.8571	***	0.0004
1 min vs. 5 min	0.3611	−0.01220 to 0.7344	ns	0.0599
2 min vs. 5 min	−0.1944	−0.3524 to −0.03652	*	0.0132

(*p* > 0.05) summarized as non-significant (ns), (*p* ≤ 0.05) as *, (*p* ≤ 0.01) as **, (*p* ≤ 0.001) as ***.

**Table 6 medicina-58-00849-t006:** Left and right control implant inter-group comparison table.

Tukey’s Multiple Comparison Test	Mean Diff.	95.00% CI of Diff.	Summary	Adjusted *p*-Value
Control				
Right Empty Lacunae vs. Left Empty Lacunae	0.6111	−0.08356 to 1.306	ns	0.1104
Right Filled Osteocytes vs. Left Filled Osteocytes	3.954	−1.229 to 9.136	ns	0.2434
Right Lacunae > 5 µm vs. Left Lacunae > 5 µm	−0.5278	−1.360 to 0.3047	ns	0.4413
Right Osteon vs. Left Osteon	−0.3333	−0.7155 to 0.04880	ns	0.1177

ns: not significant.

**Table 7 medicina-58-00849-t007:** Left and right 1 min implant inter-group comparison table.

Tukey’s Multiple Comparison Test	Mean Diff.	95.00% CI of Diff.	Summary	Adjusted *p*-Value
1 Minute				
Right Empty Lacunae vs. Left Empty Lacunae	0.04167	−0.9634 to 1.047	ns	>0.9999
Right Filled Osteocytes vs. Left Filled Osteocytes	3.472	−3.399 to 10.34	ns	0.6780
Right Lacunae > 5 µm vs. Left Lacunae > 5 µm	−1.500	−2.446 to −0.5543	***	0.0003
Right Osteon vs. Left Osteon	−0.5556	−0.9486 to −0.1625	**	0.0026

(*p* > 0.05) summarized as non-significant (ns), (*p* ≤ 0.01) as ** and (*p* ≤ 0.001) as ***.

**Table 8 medicina-58-00849-t008:** Left and right 2 min implant inter-group comparison table.

Tukey’s Multiple Comparison Test	Mean Diff.	95.00% CI of Diff.	Summary	Adjusted *p*-Value
2 Minutes				
Right Empty Lacunae vs. Left Empty Lacunae	1.264	0.2381 to 2.290	**	0.0086
Right Filled Osteocytes vs. Left Filled Osteocytes	3.694	−1.180 to 8.569	ns	0.2481
Right Lacunae > 5 µm vs. Left Lacunae > 5 µm	2.069	0.8018 to 3.337	***	0.0002
Right Osteon vs. Left Osteon	−0.05556	−0.1665 to 0.05543	ns	0.6761

(*p* > 0.05) summarized as non-significant (ns), (*p* ≤ 0.01) as ** and (*p* ≤ 0.001) as ***.

**Table 9 medicina-58-00849-t009:** Left and right 5-min implant inter-group comparison table.

Tukey’s Multiple Comparison Test	Mean Diff.	95.00% CI of Diff.	Summary	Adjusted *p*-Value
5 Minutes				
Right Empty Lacunae vs. Left Empty Lacunae	5.111	0.5614 to 9.661	*	0.0210
Right Filled Osteocytes vs. Left Filled Osteocytes	−9.236	−14.01 to −4.462	****	<0.0001
Right Lacunae > 5 µm vs. Left Lacunae > 5 µm	−0.8194	−1.887 to 0.2478	ns	0.2373
Right Osteon vs. Left Osteon	−0.2500	−0.4583 to −0.04171	*	0.0129

(*p* > 0.05) summarized as non-significant (ns), (*p* ≤ 0.05) as * and (*p* ≤ 0.0001) as ****.

**Table 10 medicina-58-00849-t010:** Periotest measurements (PTV).

	Right				Left			
Means	1 Min	2 Min	5 Min	Control	1 Min	2 Min	5 Min	Control
Implant Surgery	2.783	−6.567	−1.100	−5.767	4.100	2.967	−2.933	−2.167
Test day	−2.900	1.600	0.4000	−1.567	−2.867	0.9333	−2.200	−5.300
Day 1	0.1667	−0.1000	0.6000	2.600	−3.100	−3.200	−5.100	−5.233
Day 2	0.1667	2.567	*	0.1600	−3.100	−3.133	*	−5.200

* Implants were removed the first day after cryotherapy application.

## Data Availability

Not applicable.

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
