# Peer review of "Non-Destructive Removal of Dental Implant by Using the Cryogenic Method"

_medicina, 2022, doi:10.3390/medicina58070849_

Round 1

Reviewer 1 Report

The study seems genuine and interesting, however the authors should address the following points in the manuscript:

- Line 12 in the abstract: the following part of the sentence "In this study, it is aimed to develop" needs to be restructured.

- Line 14 in the abstract: in the following sentence " It has been tried to remove osseointegrated implants by reverse torque force method and osseo-disintegration method with cryotherapy" do authors mean two techniques were used to remove implants following osseointegration? the sentence need to be restructured.

- Line 16 in the abstract: the word "where" should be deleted and the sentence should be structured again.

- Line 26: please rephrase the following sentence: "Implants may need to be removed from the jawbone due to not being osseointegrated, not being in a suitable position, or various mechanical problems".

- The structure of this sentence is not accurate: "In primary stability, the physical shape of the implant and the structure of the bone
are effective".

- It is advisable to get an extensive English editing done on this manuscript.

*** The following comments will focus on non-linguistic issues:

- The study objectives require further clarification.

- The authors should include the proposed study hypotheses.

- The section "Implant removal procedures" needs further clarifications in particular the procedure related to this sentence " After the test day, the reverse torque application was repeated every day to examine osseodisintegration of the implants". why wasn't the removal procedure done at once?

- The authors mentioned that 8 implants were involved in the study. How were they distributed? distance between them? which ones were the tested and control sites?

- The testing sequence is somewhat confusing, please consider proper order of events.

- Line 236, what does the term "ridge loss" means?

- In the discussion section: this sentence "Most implants are osseointe-
grated and not mobile" has no true purpose or authors may consider rephrasing it.

- The following sentence in the discussion section needs a reference: " However, the success rate in this method is 87.7%".

- Please restructure the following sentence to be well-read: " The second is trephine burs. On the other hand, these burs remove the valuable intact bone from the implant's surroundings and then remove the implant".

- The discussion part is too long and presents some redundancies. You may also add study limitations and future directions for research in this field.

- The authors may summarize the conclusion section in bullet points for clarity.

Author Response

Thank you for reviewing our article.

Point-by-point response to your comments are uploaded as word file.

Reviewer 2 Report

Manuscript ID: medicina-1764497

Title: Non-Destructive Removal of Dental Implant by Cryogenic Method

1.What is the main question addressed by the research?

To develop and analyze a method for reversing the osseointegration mechanism with CO2 cryotherapy.

2.Is it relevant and interesting?

The article is relevant and interesting.

3.How original is the topic?

The topic is current.

4.What does it add to the subject area compared with other published material?

The authors have collected and analyzed a great deal of interesting data.

5.Is the paper well written?

Yes, the manuscript is well written.

6.Is the text clear and easy to read?

Yes, but minor English editing is required.

7.Are the conclusions consistent with the evidence and arguments presented?

Yes, the conclusions consistent with the evidence and arguments presented but further studies are necessary to confirm authors’ hypothesis.

8.Do they address the main question posed?

Yes, the Authors addressed the main question posed.

Other comments:

  • English language: Minor English editing is required.
  • Title: I suggest to modify the title into “can the cryogenic method be a non-destructive method for the removal of implanted fixtures? an ex vivo study”
  • Abstract: To attract the reader's attention, please clarify the target of the article, and structure the abstract.
  • Introduction: This section needs some improvements.

·       I suggest to add references after this sentence “In the removal of the implant, the reverse torque is applied first” [PMID: 32098046- doi.org/10.3390/app10238623], in particular I suggest to refer to recent literature.
Moreover, I suggest to specify what is the removal torque.

·       I suggest to add references after this sentence “In primary stability, the physical shape of the implant and the structure of the bone are effective” [PMID: 26009906 - PMID: 32475099]

·       Methods: Why did you choose the Periotest? Resonance frequency analysis could be performed in a better way with osstell or penguin.

  • Results: This section was properly prepared.
  • Discussion: This part appear too long and appears somewhere repetitive. I suggest reducing this section
  • Discussion: What is the main theme that emerges from the authors' analysis? Add an important part about study limitations? Please improve.
  • Conclusion: This section was properly prepared but further studies are necessary to confirm authors’ hypothesis.

After making the indicated changes, enriching the references part, this manuscript must to be revised.

Author Response

(The authors gave the same response as above.)

Round 2

Reviewer 1 Report

The authors have addressed the comments properly . 

Author Response

Thank you  for your  constructive and thoughtful comments.

We thank you again for the fair evaluation of our manuscript.

Reviewer 2 Report

Authors improved manuscript in several parts.

I suggest to describe in a detailed manner the usefulness of removal torque for implant stability evaluation [PMID: 32098046]

After the proper modification the manuscript must be evaluated

Author Response

Thank you for reviewing our article.

Our response to your comments are uploaded as word file.

Round 3

Reviewer 2 Report

the authors have made the suggested corrections and changes to the text to increase its quality.

I congratulate the authors